# Hypoxia Dysregulates the Transcription of Myoendothelial Junction Proteins Involved with Nitric Oxide Production in Brain Endothelial Cells

**DOI:** 10.3390/biomedicines12010075

**Published:** 2023-12-28

**Authors:** Gregory Thomas, Kaysie L. Banton, Raymond Garrett, Carlos H. Palacio, David Acuna, Robert Madayag, David Bar-Or

**Affiliations:** 1Trauma and Stroke Research Lab, 601 East Hampden Ave, Englewood, CO 80113, USA; 2Trauma and Surgery Services, Swedish Medical Center, 501 East Hampden Ave, Englewood, CO 80113, USA; 3Trauma and Surgery Services, South Texas Health System-McAllen, 301 West Expressway 83, McAllen, TX 78503, USA; 4Trauma and Surgery Services, Wesley Medical Center, 550 North Hillside St, Wichita, KS 67214, USA; 5Trauma and Surgery Services, Lutheran Medical Center, 8300 W. 38th Ave, Wheat Ridge, CO 80033, USA; 6Department of Molecular Biology, Rocky Vista University, 8401 S Chambers Rd, Parker, CO 80134, USA

**Keywords:** nitric oxide, hemoglobin alpha, endothelial nitric oxide synthase, cytochrome b5 reductase 3 myoendothelial junction

## Abstract

Myoendothelial junctions (MEJs) are structures that allow chemical signals to be transmitted between endothelial cells (ECs) and vascular smooth muscle cells, which control vascular tone. MEJs contain hemoglobin alpha (Hbα) and endothelial nitric oxide synthase (eNOS) complexes that appear to control the production and scavenging of nitric oxide (NO) along with the activity of cytochrome b5 reductase 3 (CYB5R3). The aim of this study was to examine how hypoxia affected the regulation of proteins involved in the production of NO in brain ECs. In brief, human brain microvascular endothelial cells (HBMEC) were exposed to cobalt chloride (CoCl_2_), a hypoxia mimetic, and a transcriptional analysis was performed using primers for *eNOS*, *CYB5R3*, and *Hbα2* with ΔΔCt relative gene expression normalized to *GAPDH*. NO production was also measured after treatment using 4,5-diaminofluorescein diacetate (DAF-DA), a fluorescent NO indicator. When HBMEC were exposed to CoCl_2_ for 48 h, *eNOS* and *CYB5R3* messenger RNA significantly decreased (up to −17.8 ± 4.30-fold and −10.4 ± 2.8, respectively) while *Hbα2* increased to detectable levels. Furthermore, CoCl_2_ treatment caused a redistribution of peripheral membrane-generated NO production to a perinuclear region. To the best of our knowledge, this is the first time this axis has been studied in brain ECs and these findings imply that hypoxia may cause dysregulation of proteins that regulate NO production in brain MEJs.

## 1. Introduction

Nitric oxide (NO) is a gaseous signaling molecule essential for proper endothelial function and the maintenance of vascular tone [1]. However, due to its highly reactive nature, the cellular location of its production must be tightly controlled to prevent off-target interactions. Thus, structures known as myoendothelial junctions (MEJs) have evolved to promote NO synthesis by endothelial cells (ECs) proximal to vascular smooth muscle cells (VSMC) within resistance arteries [2]. Moreover, MEJs act as the center of a calcium (Ca^2+^) and NO feedback loop that helps control vascular tone [3].

The dynamic production of NO by ECs within MEJs is driven by the activity of nitric oxide synthase (eNOS or NOS3) [4]. Under resting conditions, eNOS is found complexed with caveolin-1 at the plasma membrane, placing it in an inhibitory state [4]. On the other hand, increases in intracellular calcium facilitate the recruitment of cofactors which release eNOS to generate NO from the amino acid L-arginine via an oxygen reduction reaction [4]. NO can then freely diffuse into nearby VSMCs, activating soluble guanylate cyclase and relaxing the vessel wall [4]. The equilibrium between NO synthesis and bioavailability is therefore a key factor in vascular tone.

Recent research suggests that the overall level of NO produced in MEJs is determined via redox reactions involving the iron present in hemoglobin α (Hbα). Unpredictably, ECs express Hbα, where it has been found complexed with eNOS within MEJs and acts to control NO diffusion based on the oxidation state of the heme iron present and the availability of oxygen [3]. In the presence of oxygen, the ferrous (Fe^2+^)/Hbα complexes scavenge NO via a dioxygenase reaction, in which NO rapidly reacts with O_2_-bound (Fe^2+^)/Hbα to form nitrate (NO_3_^−^) and ferric (Fe^3+^)/Hbα [3]. Cytochrome b5 reductase 3 (CYB5R3), another enzyme found in the MEJ, then reduces the iron back to a ferrous state to repeat the process [2]. The ferric (Fe^3+^)/Hbα complex, on the other hand, reacts slowly with NO to preserve diffusion into VSMC [3]. Animal studies also show that under hypoxic conditions, Hbα residing in the vessel wall can act as a nitrate reductase to provide NO in the absence of oxygen [5]. As a result, NO regulation in MEJs is a complex and multifaceted process (Figure 1).

Many chronic and acute conditions, such as hypertension, atherosclerosis, stroke, coronary artery disease, and peripheral artery disease, are thought to be exacerbated by a loss of control over vascular tone [6]. For instance, the increased vasoconstriction observed in hypertensive patients leads to elevated resistance and blood pressure which, if untreated, can lead to vessel and organ damage [6,7]. Furthermore, cerebral vasospasm (CV) is a significant challenge in the management of patients following subarachnoid hemorrhage (SAH) and has been linked to delayed ischemic neurological deficits [8].

In this investigation, we hypothesized that hypoxia affects the transcriptional regulation of proteins involved in NO production within MEJs. We discovered that activating hypoxia-induced factor-1α (HIF-1α) in brain microvascular vascular endothelial cells with cobalt chloride increases *Hbα* transcription while decreasing *eNOS* and *CYB5R3*. These findings suggest that hypoxia-induced activation of HIF-1α may influence both the overall production of NO by brain endothelial cells as well as how it is regulated in MEJs. Understanding the mechanisms underlying this axis could lead to the development of new treatments for vasospasms and brain injuries.

## 2. Materials and Methods

### 2.1. Materials

Cell culture reagents and 4,5-diaminofluorescin diacetate (DAF-FM acetate) were purchased from ThermoFisher Scientific (Waltham, MA, USA). Primary human brain microvascular endothelial cells (HBMEC) were obtained from Cell systems (Kirkland, WA, USA) and endothelial cell growth medium MV2 (EGM-MV2) from Promocell (Heidelberg, Germany). IL-1β and TNFα were purchased from R&D systems (Minneapolis, MN, USA). All other reagents were obtained from MilliporeSigma (St. Louis, MO, USA).

### 2.2. Cell Culture, Chemically Induced Hypoxia, and NO Detection

Passage 8 HBMEC were seeded into culture plates at 62,500 cells/cm^2^ in EGM-MV2 and incubated for up to 48 h at 37 °C and 5% CO_2_. Working solutions of CoCl_2_, IL-1β, and TNFα prepared in Hank’s balanced salt solution (HBSS) were then added to final concentrations of 100, 200, or 500 μM of CoCl_2_ or 100 ng/mL of cytokine, and the cells were cultured for an additional 48 h. Total RNA was then isolated using Qiagen RNeasy Plus columns (Germantown, MD, USA) and real-time PCR performed for glyceraldehyde-3-phosphate dehydrogenase (GAPDH), nitric oxide synthase 3 (eNOS), cytochrome b5 reductase 3 (CYB5R3), or hemoglobin alpha 2 (HBA2) using validated primers and reagents obtained from Qiagen. Following amplification, transcriptional analysis was performed using ΔCt analysis of threshold call differences between GAPDH and targets, or relative expression calculated by ΔΔCt analysis versus diluent controls. For post-treatment detection of NO, some wells were loaded with 5 μM DAF-FM acetate for one hour at room temperature and washed, and NO production was fluorescently measured after 2 h in HBSS using a Nikon Eclipse Ti microscope (Melville, NY, USA).

### 2.3. Statistics

Two-tailed, one-sample *t*-tests (hypothetical value = 1; α = 0.05) were used to determine significant fold-changes in relative expression, and Grubb’s test was used to identify outliers with the Real Statistics Resource Pack (https://real-statistics.com/free-download/real-statistics-resource-pack/).

## 3. Results

### 3.1. CoCl_2_ Reduces the Transcription of eNOS and CYB5R3 in HBMEC

HBMECs were treated with CoCl_2_ to assess the effects of HIF-1α signaling, a major pathway activated during hypoxia, on proteins involved with the production and regulation of NO by endothelial cells [9]. Fold changes in *eNOS*, *CYB5R3*, or *HBA2* transcription were then evaluated using ΔΔCt relative gene expression in isolated RNA samples with normalization against *GAPDH*.

Initially, a 2- to 48-h temporal evaluation of CoCl_2_ treatment was performed with final concentrations ranging from 100 to 1000 μM. We found that the targets of interest experienced dose- and time-dependent changes as well as toxicity at concentrations greater than 500 μM. As a result of the pronounced dynamic range of the response, 48-h treatments, with a maximum dose of 500 μM, were chosen for final statistical analysis. At this time point, CoCl_2_ treatment of HBMEC resulted in a dose-dependent reduction in the detectable amounts of both *eNOS* and *CYB5R3* messenger RNA levels (Figure 2A; *n* = 5 for 500 μM CoCl_2_ and *n* = 6 for all other treatment groups). For 100, 200, and 500 μM final concentrations, the observed mean of means ± SEM fold change reduction in eNOS was −3.1 ± 0.8, −6.9 ± 1.6, and −17.8 ± 4.30, respectively (*p*-values ≤ 0.05). A similar response for *CYB5R3* was observed with fold change reductions of −1.4 ± 0.5, −3.2 ± 0.5, and −10.4 ± 2.8, respectively (*p*-values ≤ 0.05). These findings show that hypoxia reduces critical enzymes involved in the balance of NO produced by HBMEC.

### 3.2. CoCl_2_-Induced eNOS and CYB5R3 Transcriptional Programs Differ from Inflammatory Programs

To determine if hypoxia-induced transcriptional changes mimic those derived by an inflammatory milieu, HBMEC were treated with 100 ng/mL final concentrations of IL-1β or TNFα for 48 h and relative gene expression of *eNOS* and *CYB5R3* were compared to CoCl_2_ treatment (Figure 2B; *n* = 5 for 500 μM CoCl_2_ and IL-1β or *n* = 4 for TNFα). These cytokine concentrations are higher than found physiologically [10,11] and represent a saturating cytokine stimulation. All treatment groups reduced *eNOS* after 48 h; however, higher fold changes were observed under hypoxic conditions versus cytokine exposure, or −17.8 ± 4.30 for 500 μM CoCl_2_ compared to −4.0 ± 1.5 for IL-1β and 4.1 + 0.8 for TNFα (*p*-values ≤ 0.05). Importantly, exposure to inflammatory cytokines did not affect *CYB5R3* fold-regulation. These data indicate that hypoxic transcriptional programs for these targets differ from those induced by inflammation.

### 3.3. CoCl_2_ Induces the Transcription of HBA2 in HBMEC

In contrast, hypoxia appears to induce *HBA2* transcription in HBMEC, with a dose-dependent increase in the amount of detectable messenger RNA observed after 48 h. Figure 3 depicts representative amplification curves demonstrating that, while GAPDH cycle calls for CoCl_2_ dose responses remain consistent, detectable levels of *HBA2* appear to increase in correlation with dose. It is also worth noting that *HBA2* was found to be a rare transcript in our model for resting cells, which prevented both accurate amplification and relative expression calculation versus diluent controls. To support this observation, ΔCt amplification call measurements for *HBA2* to *GAPDH* were calculated for the corresponding experiment shown in Figure 3, which was performed in triplicate. Using this method, we observed ΔCts of 26.1 ± 0.4, 24.6 ± 0.8, and 21.8 ± 0.2 for 100, 200, and 500 μM final concentrations of CoCl_2_, respectively. This would imply that as the concentration of CoCl_2_ increases, so does the detectable amount of *HBA2* messenger RNA. Furthermore, as seen in the diluent controls, cytokine treatment did not result in detectable *HBA2* amplification (none detected). Together, these findings suggest that HIF-1 signaling increases Hbα transcription.

### 3.4. CoCl_2_ Induces Changes in the Localization of NO Production

To investigate the functional implications of exposure to CoCl_2_, HBMEC cells were loaded with a fluorescent NO indicator following treatment, and NO production was measured after two hours under an epifluorescent microscope. After 48 h of treatment, we observed changes in the localization of NO within HBMEC. Cells exposed to diluent control appear to produce diffuse amounts of NO around the cell’s periphery (Figure 4A), whereas those exposed to 500 μM CoCl_2_ appear to produce elevated levels of perinuclear NO (Figure 4B). This would imply that, while NO synthesis may be internally enhanced, it is not occurring at exterior/plasma membrane locations under hypoxic conditions.

## 4. Discussion

In this report, we show that mimicking hypoxia in HBMEC, by activating HIF-1α with cobalt chloride, influences the transcription of MEJ proteins in ways that could drive vascular pathology. One of our key findings was that hypoxia inhibits the transcription of eNOS, the primary source of NO in the MEJ. Surprisingly, there appears to be some disparity in the literature for this response. Hypoxia-induced microRNAs, for example, have been shown to downregulate eNOS and NO bioavailability by reducing the half-life of *eNOS* mRNA in umbilical vein endothelial cells [12]. On the other hand, hypoxia has been shown to upregulate cAMP-mediated pathways that promote the expression of eNOS and enhance NO production in bovine endothelial cells [13]. Despite these discrepancies, it is easy to speculate that the reduced transcription of *eNOS* observed in our experiments may limit the production of NO at the MEJ.

Our findings also indicate that hypoxia activates compensatory transcriptional programs designed to protect NO. In the first of these mechanisms, we observed a dose-dependent reduction in CYB5R3, which, as previously stated, is an enzyme that serves to reduce the iron within the eNOS/Hbα complex to a NO scavenging Fe^2+^ state [2]. In support of this, Straub et al. elegantly demonstrated that eNOS/Hbα-derived NO helps regulate coronary vascular tone and that knockdown of CYB5R3 decreases arterial reactivity [14]. Interestingly, loss of CYB5R3 expression in VSMC has been shown to lower soluble guanylate cyclase (sGC) and cGMP levels [15]. Thus, reduced CYB5R3 expression may skew the balance of the iron present to Fe^3+^, effectively chelating the remaining NO present and promoting slow release across the MEJ during the hypoxic event.

Endothelial cells could also stimulate NO production in the absence of oxygen by increasing Hbα expression. Interestingly, we observed that CoCl_2_ treatment of HBMEC promotes the transcription of *Hbα.* This is significant because recent evidence suggests that during hypoxia, Hbα functions as a nitrite reductase. Keller et al. demonstrated in mouse models that vessels isolated from wild-type animals consumed nitrate in response to hypoxia, resulting in vasodilation, whereas this response was absent in mice edited using CRISPR/Cas9 to express a form of Hbα that prevents eNOS binding [5]. Moreover, they found that these edited mice also exhibit significantly increased NO production under normoxic conditions [5], indicating a potential consequence of Hbα overexpression in the presence of oxygen. Humans appear to have a similar predisposition in α-thalassemia, a genetic hemoglobinopathy caused by the loss of Hbα, which is associated with increased NO-induced vascular perfusion [16]. Furthermore, in vivo studies using a Hbα mimetic peptide have shown that competitively uncoupling Hbα from eNOS results in increased NO signaling in the vessel wall, decreased alpha-1-adrenergic vasoconstriction, lower mean arterial blood pressure, and protection against angiotensin-2 induced hypertension in mice [17,18]. Taken together, Hbα is important in modulating vascular function, but upregulation may be detrimental to NO levels during reperfusion.

It should be noted that this study had limitations. To begin, the cells were not co-cultured with VSMC, which could have provided important cell-to-cell interactions. Second, total protein was not examined, which would have revealed information about translation and localization. Having said that, all the documented transcriptional changes in this report derive from endothelial cells and are likely to affect protein levels and regulatory dynamics in MEJs. In addition, we are unsure how the doses and exposure times of CoCl_2_ used in these experiments relate clinically or physiologically. In response, we believe that the dose-dependence of the observed responses strengthens the validity of our findings and that in vivo local microenvironments are subjected to hypoxic conditions for these durations. However, we believe that the findings show that hypoxia may activate transcriptional programs in brain endothelial cells, which may contribute to vasospasm pathology. Furthermore, additional research into the regulation of this axis by HIF-1 may aid in the development of therapies for the treatment of brain injuries characterized by vascular tone dysregulation.

## 5. Conclusions

In conclusion, these findings suggest that hypoxia may activate transcriptional programs that contribute to vasospasm pathology. HIF-1α signaling appears to reduce eNOS transcription, which may lower the overall protein level of a protein required for NO production. Furthermore, transcriptional programs are triggered to compensate for this loss. CYB5R3 transcription, for example, is reduced to potentially prevent the formation of NO-scavenging Hbα complexes within MEJs. Conversely, Hbα transcription is upregulated to protect the remaining NO while also producing NO from alternate sources. However, during reperfusion, these transcriptional programs may promote excessive NO scavenging. To the best of our knowledge, this axis has never been studied in brain ECs, and future research into these pathways could lead to novel therapies for treating local or anoxic injuries caused by brain injury or edema.

## Figures and Tables

**Figure 1 biomedicines-12-00075-f001:**
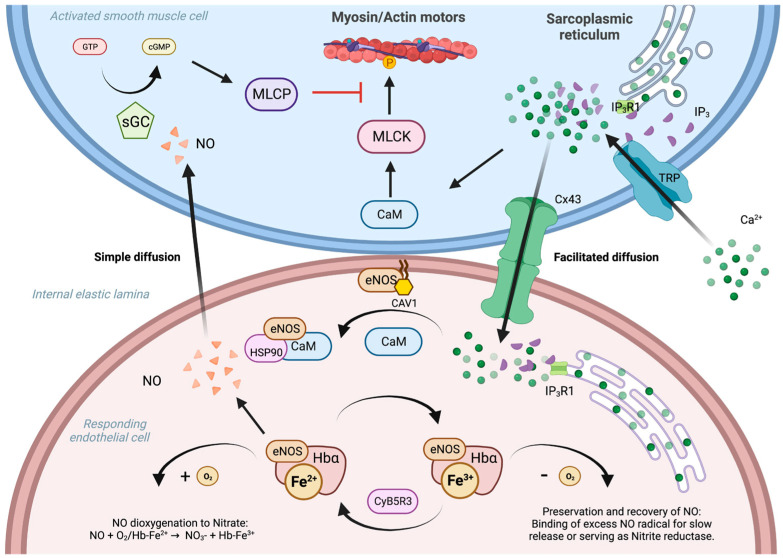
Feedback mechanisms controlling the regulation of NO in the MEJ. Endothelial nitric oxide synthase (eNOS) production within the myoendothelial junction (MEJ) is triggered in response to Ca^2+^ derived from the connected vascular smooth muscle cell (VSMC). The process begins when Inositol 1,4,5-triphosphate (IP_3_) produced by activated VSMC binds to IP_3_-receptor 1 (IP_3_R1) on the sarcoplasmic reticulum or transient potential receptor (TRP) at the membrane to increase cytosolic Ca^2+^ concentrations. This increase results in the phosphorylation of actin/myosin motors via a calmodulin (CaM)/myosin light chain kinase (MLCK) cascade. Concurrently, IP_3_ opens connexin 43 (Cx43) channels to allow for an influx of Ca^2+^ and IP_3_ into the tethered endothelial cell (EC). The resulting activated CaM then binds with eNOS and heat shock protein 90 (HSP90) to form an active complex that is released from caveolin 1 (CAV1). Subsequently produced NO can then diffuse back into VSMCs, where it activates soluble guanylyl cyclase (sGC), facilitating the conversion of GTP to cGMP, which ultimately leads to a reduction in constriction by myosin light chain phosphatase (MLCP). In the presence of oxygen, NO can be scavenged via deoxygenation reactions catalyzed by hemoglobin α (Hbα)/Fe^2+^/eNOS complexes to form both nitrate (NO_3_^−^) and oxidized Hbα/Fe^3+^ complexes. This reaction is potentiated by the reduction of Hbα/Fe^3+^ back to Hbα/Fe^2+^ by cytochrome B5 reductase 3 (CyBR3). In the absence of oxygen, Hbα/Fe^3+^ can bind NO, to preserve its activity, or serve as a nitrite reductase. Figure created with Biorender.com.

**Figure 2 biomedicines-12-00075-f002:**
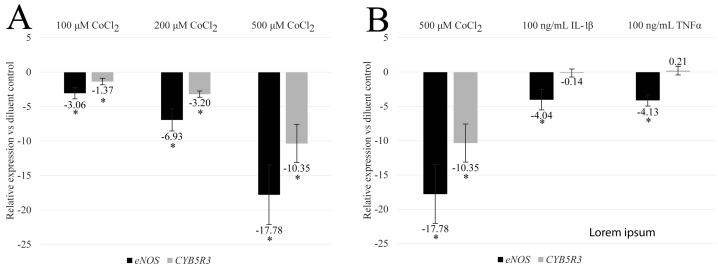
CoCl_2_ dose-dependently inhibits *eNOS* and *CYB5R3* transcription in HBMEC. To evaluate the effects of hypoxia on the transcription of *eNOS* and *CYB5R3*, total RNA was isolated from human brain microvascular endothelial cells treated with increasing doses of CoCl_2_ for 48 h. ΔΔCt relative gene expression was then calculated for *eNOS* and *CYB5R3* and normalized to *GAPDH*. Data presented as mean of means ± SEM fold change in expression for 48-h CoCl_2_ dose-responses (**A**) or 48-h 100 ng/mL IL-1β or TNFα compared to 500 μM CoCl_2_ (**B**). * = *p*-value ≤ 0.05 vs. activated saline control by one sample *t*-test (hypothetical value = 1; α = 0.05; *n* = 4 for TNFα, *n* = 5 for 500 μM CoCl_2_ and IL-1β, and *n* = 6 for all other treatment groups).

**Figure 3 biomedicines-12-00075-f003:**
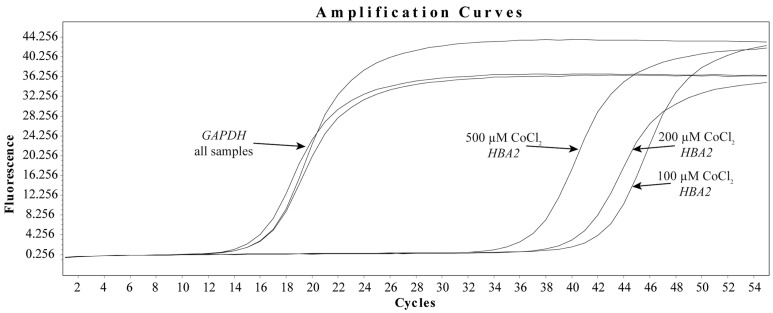
CoCl_2_ dose-dependently increases *Hbα* transcription in HBMEC. To evaluate the effects of hypoxia on the transcription of *Hbα*, total RNA was isolated from human brain microvascular endothelial cells treated with increasing doses of CoCl_2_ for 48 h. qRT-PCR was performed using specific primers for *Hbα* and *GAPDH*. Data presented as representative amplification curves.

**Figure 4 biomedicines-12-00075-f004:**
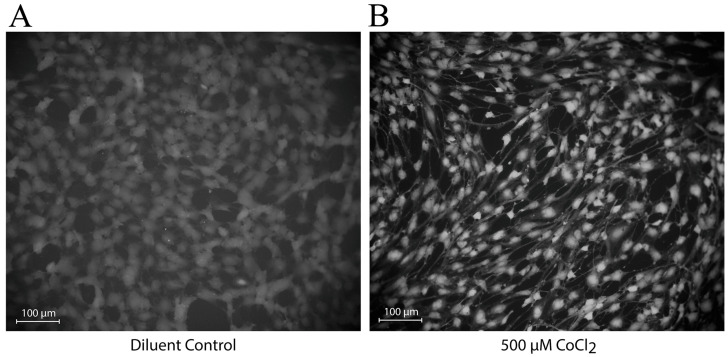
CoCl_2_ treatment of HBMEC alters the location of NO production. To evaluate the effects of hypoxia on the production of NO, human brain microvascular endothelial cells were treated with a final concentration of CoCl_2_ for 48 h and then loaded with a fluorescent NO indicator (DAF-FM acetate). Overall NO was measured under an epifluorescent microscope after two hours of development at room temperature: 48-h diluent control (**A**) and 48-h 500 μM CoCl_2_ (**B**).

## Data Availability

Data are contained within the article.

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
