# Peer review of "Hypoxia Dysregulates the Transcription of Myoendothelial Junction Proteins Involved with Nitric Oxide Production in Brain Endothelial Cells"

_biomedicines, 2023, doi:10.3390/biomedicines12010075_

Round 1
Reviewer 1 Report
Comments and Suggestions for Authors
This article is comprehensive, logically organized, and contains valuable information on hypoxia dysregulates the transcription of myoendothelial junction proteins involved with nitric oxide production in brain endothelial cells. The authors did excellent research on investigating how hypoxia affected the regulation of proteins involved in the production of NO in brain endothelial cells. The authors demonstrated that hypoxia-induced factor-1a signaling appears to reduce endothelial nitric oxide synthase transcription, which may lower the overall protein level of a protein required for NO production.
To improve the manuscript, the authors should take the following considerations:
(1) The authors presented that the CoCl2 dose-dependently increases Hba transcription in HBMEC in Figure 3. The authors mentioned that the GAPDH cycle calls for CoCl2 dose responses to remain consistent. Why the threshold points are not the same for all samples of GAPDH? The authors should calculate the qRT-PCR final fluorescent intensity in terms of Accuracy (%), Sensitivity (%), and Specificity (%) to prove the detectable levels of HBA2 appear to increase in correlation with dose. The authors mentioned, “Cytokine treatment was found to have no impact on the transcription of HBA2 (data not shown).” It is suggested that the authors should show the cytokine treatment data on the transcription of HBA2 to improve the manuscript.
(2) The authors presented that the CoCl2 treatment of HBMEC alters the location of NO production in Figure 4. The authors should include the epifluorescent microscope after two hours of development at room temperature for 100 and 200 µM CoCl2 to gain a better understanding of the CoCl2 treatment of HBMEC alters the location of NO production.
The submitted manuscript has significant scientific insights and the experimental data support the conclusions. However, the present submission requires minor revisions before being considered for publication in the well-circulated Biomedicines in its current condition.
Author Response
For research article biomedicines-2782826
“Hypoxia dysregulates the transcription of myoendothelial junction proteins involved with nitric oxide production in brain endothelial cells”
Response to Reviewer #1
We express our gratitude to Reviewer #1 for critically evaluating our manuscript. Please refer to our detailed responses below, along with the corresponding edits, corrections, and tracked changes that have been re-submitted.
General Questions for evaluation
Does the introduction provide sufficient background and include all relevant references? Yes
Response: NA
Are all the cited references relevant to the research? Yes
Response: NA
Is the research design appropriate? Yes
Response: NA
Are the methods adequately described? Can be improved.
Response:
We thank reviewer #1 for their critical review of our work and for bringing to our attention that the methods section does not adequately describe the relevant methodologies used. To address this oversight, revisions have been made, which will be detailed in our itemized, point-by-point response to the reviewer's comments below.
Are the results clearly presented? Can be improved.
Response:
We would like to thank reviewer #1 once more for their attention to detail and for pointing out that our results section could be improved for clarity. To correct this, the appropriate revisions to reviewer #1’s point-driven comments are provided below.
Are the conclusions supported by the results? Yes
Response: NA
Point-by-point response to Comments and Suggestions for Authors
This article is comprehensive, logically organized, and contains valuable information on hypoxia dysregulates the transcription of myoendothelial junction proteins involved with nitric oxide production in brain endothelial cells. The authors did excellent research on investigating how hypoxia affected the regulation of proteins involved in the production of NO in brain endothelial cells. The authors demonstrated that hypoxia-induced factor-1a signaling appears to reduce endothelial nitric oxide synthase transcription, which may lower the overall protein level of a protein required for NO production.
To improve the manuscript, the authors should take the following considerations:
Comment #1
- The authors presented that the CoCl2dose-dependently increases Hba transcription in HBMEC in Figure 3. The authors mentioned that the GAPDH cycle calls for CoCl2 dose responses to remain consistent. Why the threshold points are not the same for all samples of GAPDH? The authors should calculate the qRT-PCR final fluorescent intensity in terms of Accuracy (%), Sensitivity (%), and Specificity (%) to prove the detectable levels of HBA2 appear to increase in correlation with dose.
Response:
We agree with reviewer #1 that the results presented in Figure 3 could be more clearly presented and the methods described more comprehensively. However, the qRT-PCR transcriptional analysis for this study did not use absolute quantification or standard curves. As a result, we are unable to provide the metrics (accuracy, sensitivity, and specificity) suggested by reviewer #1 using the data generated. Having said that, we believe that reviewer #1 is correct in stating that measurables will provide stronger evidence for dose dependency. We compiled delta Ct measurements or threshold call differences between GAPDH and HBA2 for the companion experiments (n=3) presented in Figure 3 to provide such a measurement. By this method, we observed delta Cts of 26.1 + 0.4, 24.6 + 0.8, and 21.8 + 0.2 for 100, 200, and 500 mM final concentrations of CoCl2 in our model, respectively. This suggests that as CoCl2 concentration increases, so does the detectable amount of HBA2 messenger RNA to housekeeping gene amplification. The following has been added at line 181 to both relay and clarify these findings.
“To support this observation, DCt amplification call measurements for HBA2 to GAPDH were calculated for the corresponding experiment shown in Figure 3, which was performed in triplicate. Using this method, we observed DCts of 26.1 + 0.4, 24.6 + 0.8, and 21.8 + 0.2 for 100, 200, and 500 mM final concentrations of CoCl2, respectively. This would imply that as the concentration of CoCl2 increases, so does the detectable amount of HBA2 messenger RNA.”
This comment, we believe, also complements reviewer #1’s general question comment for improving the methods descriptions. We have added the following to the methods section at line 116 to clarify that the method used is relative to controls and was performed using validated primers obtained from a reputable source (Qiagen), as well as relay that we have included new metrics and tests.
From:
“Total RNA was then isolated using Qiagen RNeasy plus columns (Germantown, MD) and quantitative real-time PCR performed for Glyceraldehyde-3-phosphate dehydrogenase (GAPDH), Nitric oxide synthase 3 (eNOS), Cytochrome b5 reductase 3 (CYB5R3), or Hemoglobin alpha 2 (HBA2) using primers and reagents obtained from Qiagen.“
To:
“Total RNA was then isolated using Qiagen RNeasy plus columns (Germantown, MD) and real-time PCR performed for Glyceraldehyde-3-phosphate dehydrogenase (GAPDH), Nitric oxide synthase 3 (eNOS), Cytochrome b5 reductase 3 (CYB5R3), or Hemoglobin alpha 2 (HBA2) using validated primers and reagents obtained from Qiagen. Following amplification, transcriptional analysis was performed using DCt analysis of threshold call differences between GAPDH and targets, or relative expression calculated by DDCt analysis versus diluent controls.”
As for the threshold points not being the same for all samples, single representative amplification curves are presented in Figure 3. Single curves were randomly selected from the plate to visually demonstrate our findings for increased CoCl2-induced HBA2 transcription in our model but keep the figure from being too “busy”. As a result, select curves will be subject to small degrees of uncontrollable variance stemming from factors such as pipetting errors or potential location bias across the plates that will be observed as small threshold differences. The delta Ct and delta-delta analysis employed corrects for these variances, to give normalized values for final analysis. We feel that based on reviewer #1’s suggestions, our paper has been enhanced to both provide high-level, graphical representation (.i.e. Figure 3) that is now supported by the new delta Ct findings being included in the revision.
Comment #2
The authors mentioned, “Cytokine treatment was found to have no impact on the transcription of HBA2 (data not shown).” It is suggested that the authors should show the cytokine treatment data on the transcription of HBA2 to improve the manuscript.
Response:
We would like to thank reviewer #1 once more for their insightful suggestions. Based on this feedback, we believe that the noted passage does not adequately convey the lack of amplification for HBA2 observed in our model and, while we cannot provide data for the absence of amplification, the following sentence at line 186 has been changed as follows.
From:
“In addition, cytokine treatment was found to have no impact on the transcription of HBA2 (data not shown).”
“
To:
“Furthermore, as seen in the diluent controls, cytokine treatment did not result in detectable HBA2 amplification (none detected).”
Comment #3
(2) The authors presented that the CoCl2 treatment of HBMEC alters the location of NO production in Figure 4. The authors should include the epifluorescent microscope after two hours of development at room temperature for 100 and 200 µM CoCl2 to gain a better understanding of the CoCl2 treatment of HBMEC alters the location of NO production.
Response:
We agree with reviewer #1 that additional epifluorescent images, at different concentrations, could aid in a better understanding of these suspected localization changes. We are currently working on compiling such images, as well as more directed localization experiments of the proteins studied in this investigation. We will be unable to provide additional images for resubmission due to the time constraints of the manuscript revisions. However, we believe that these will serve as the foundation for further research in this exciting project.
Reviewer 2 Report
Comments and Suggestions for Authors
The paper is focused on the influence of hypoxia on the regulation of myoendothelial junction proteins involved in the production of NO in brain ECs. The topic could be interesting for readers of Biomedicine. The presentation and discussion of the results are accurate and correct. I recommend the publication after the following minor revisions:
- Fig. 4. Please report the scale length within the images.
- Conclusions and Abstract should be improved by evidencing the novelty of this work with respect to literature.
Comments on the Quality of English LanguageMinor corrections are needed.
Author Response
For research article biomedicines-2782826
“Hypoxia dysregulates the transcription of myoendothelial junction proteins involved with nitric oxide production in brain endothelial cells”
Response to Reviewer #2
We would also like to thank Reviewer #2 for critically assessing our manuscript. Please refer to our detailed responses below, along with the corresponding edits, corrections, and tracked changes that have been re-submitted.
General Questions for evaluation
Does the introduction provide sufficient background and include all relevant references? Yes
Response: NA
Are all the cited references relevant to the research? Yes
Response: NA
Is the research design appropriate? Yes
Response: NA
Are the methods adequately described? Yes
Response:
Are the results clearly presented? Can be improved.
Response: We would like to thank reviewer #2 for pointing out that our results section could be concise. This, we believe, has been addressed as part of the same comment provided by reviewer #1. Please see the detailed responses above.
Are the conclusions supported by the results? Can be improved
Response:
We appreciate reviewer #2’s critical assessment of our work and for bringing to our attention that our conclusion statement could be improved. Revisions have been made to address this oversight, as detailed in our itemized, point-by-point response to the reviewer's comments below.
Point-by-point response to Comments and Suggestions for Authors
The paper is focused on the influence of hypoxia on the regulation of myoendothelial junction proteins involved in the production of NO in brain ECs. The topic could be interesting for readers of Biomedicine. The presentation and discussion of the results are accurate and correct. I recommend the publication after the following minor revisions:
Comment #1
- Fig. 4. Please report the scale length within the images.
Response:
We are grateful to reviewer #2 for allowing us to correct this oversight. In the revised manuscript, Figure 4 has been updated to include scale length bars.
- Conclusions and Abstract should be improved by evidencing the novelty of this work with respect to literature.
Response:
We agree with reviewer #2 that the novelty of our work should be highlighted more prominently in both the abstract and conclusion. To address this, we have added the following:
The last two sentences of the abstract, at line 30 have been revised as follows:
From:
“These findings imply that hypoxia may cause dysregulation of proteins that regulate NO production in MEJs. Furthermore, NO suppression and/or increased scavenging may contribute to the pathology of conditions characterized by loss of vascular tone control.”
To:
“To the best of our knowledge, this is the first time this axis has been studied in brain ECs and these findings imply that hypoxia may cause dysregulation of proteins that regulate NO production in brain MEJs.”
Also, the last sentence of the conclusion, at line 274, has been changed as follows:
From:
“Further research into these pathways could lead to novel therapies for treating local or anoxic injuries caused by brain injury or edema.”
To:
“To the best of our knowledge, this axis has never been studied in brain ECs, and future research into these pathways could lead to novel therapies for treating local or anoxic injuries caused by brain injury or edema.”
Response to Comments on the Quality of English Language
We would like to take one last opportunity to thank Reviewer #2 for thoroughly reviewing our work and pointing out that there are opportunities to improve the language. To address this, additional revisions have been incorporated and highlighted in the revised manuscript.